# ITERATIVE PATCH SELECTION FOR HIGH-RESOLUTION IMAGE RECOGNITION

**Benjamin Bergner**[1], **Christoph Lippert**[1,2], **Aravindh Mahendran**[3]
[1]Hasso Plattner Institute for Digital Engineering, University of Potsdam
[2]Hasso Plattner Institute for Digital Health at the Icahn School of Medicine at Mount Sinai
[3]Google Research, Brain Team
`{firstname.lastname}@hpi.de  aravindhm@google.com`

## ABSTRACT

High-resolution images are prevalent in various applications, such as autonomous driving and computer-aided diagnosis. However, training neural networks on such images is computationally challenging and easily leads to out-of-memory errors even on modern GPUs. We propose a simple method, Iterative Patch Selection (IPS), which decouples the memory usage from the input size and thus enables the processing of arbitrarily large images under tight hardware constraints. IPS achieves this by selecting only the most salient patches, which are then aggregated into a global representation for image recognition. For both patch selection and aggregation, a cross-attention based transformer is introduced, which exhibits a close connection to Multiple Instance Learning. Our method demonstrates strong performance and has wide applicability across different domains, training regimes and image sizes while using minimal accelerator memory. For example, we are able to finetune our model on whole-slide images consisting of up to 250k patches (>16 gigapixels) with only 5 GB of GPU VRAM at a batch size of 16.

## 1 INTRODUCTION

Image recognition has made great strides in recent years, spawning landmark architectures such as AlexNet (Krizhevsky et al., 2012) or ResNet (He et al., 2016). These networks are typically designed and optimized for datasets like ImageNet (Russakovsky et al., 2015), which consist of natural images well below one megapixel.[1] In contrast, real-world applications often rely on high-resolution images that reveal detailed information about an object of interest. For example, in self-driving cars, megapixel images are beneficial to recognize distant traffic signs far in advance and react in time (Sahin, 2019). In medical imaging, a pathology diagnosis system has to process gigapixel microscope slides to recognize cancer cells, as illustrated in Fig. 1.

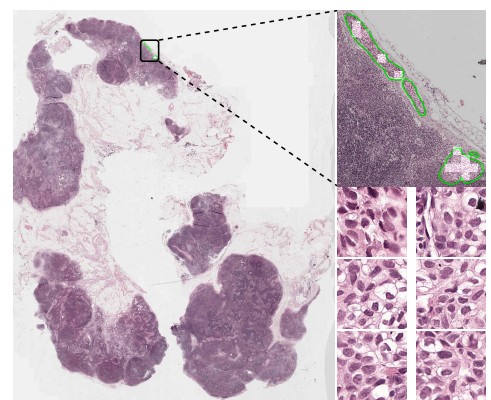

Figure 1: IPS selects salient patches for high-resolution image recognition. In the top right insert, green contours represent ground truth cancerous cells and white overlays indicate high scoring patches from our model. Example patches are shown in the bottom right.

Training neural networks on high-resolution images is challenging and can lead to out-of-memory errors even on dedicated high-performance hardware. Although downsizing the image can fix this problem, details critical for recognition may be lost in the process (Sabottke & Spieler, 2020; Katharopoulos & Fleuret, 2019). Reducing the batch size is another common approach to decrease memory usage, but it does not scale to arbitrarily large inputs and may lead to instabilities in networks involving batch normalization (Lian & Liu, 2019). On the other hand, distributed learning across multiple devices increases resources but is more costly and incurs higher energy consumption (Strubell et al., 2019).

---

[1]For instance, a 256×256 image corresponds to only 0.06 megapixels.

We propose Iterative Patch Selection (IPS), a simple patch-based approach that decouples the consumed memory from the input size and thus enables the efficient processing of high-resolution images without running out of memory. IPS works in two steps: First, the most salient patches of an image are identified in no-gradient mode. Then, only selected patches are aggregated to train the network. We find that the attention scores of a cross-attention based transformer link both of these steps, and have a close connection to Multiple Instance Learning (MIL).

In the experiments, we demonstrate strong performance across three very different domains and training regimes: traffic sign recognition on megapixel images, multi-task classification on synthetic megapixel MNIST digits, and using self-supervised pre-training together with our method for memory-efficient learning on the gigapixel CAMELYON16 benchmark. Furthermore, our method exhibits a significantly lower memory consumption compared to various baselines. For example, when scaling megapixel MNIST images from 1k to 10k pixels per side at a batch size of 16, we can keep peak memory usage at a constant 1.7 GB while maintaining high accuracy, in contrast to a comparable CNN, which already consumes 24.6 GB at a resolution of 2k×2k. In an ablation study, we further analyze and provide insights into the key factors driving computational efficiency in IPS. Finally, we visualize exemplary attention distributions and present an approach to obtain patch-level class probabilities in a weakly-supervised multi-label classification setting.

## 2 Methods

We regard an image as a set of $N$ patches. Each patch is embedded independently by a shared encoder network, resulting in $D$-dimensional representations, $\boldsymbol{X} \in \mathbb{R}^{N \times D}$. Given the embeddings, we select the most salient patches and aggregate the information across these patches for the classification task. Thus our method, illustrated in Fig. 2, consists of two consecutively executed modules: an iterative **patch selection** module that selects a fixed number of patches and a transformer-based **patch aggregation** module that combines patch embeddings to compute a global image embedding that is passed on to a classification head. Crucially, the patch aggregation module consists of a cross-attention layer, that is used by the patch selection module in no-gradient mode to score patches. We discuss these in detail next and provide code at `https://github.com/benbergner/ips`.

### 2.1 Iterative Patch Selection

Given infinite memory, one could use an attention module to score each patch and select the top $M$ patches for aggregation. However, due to limited GPU memory, one cannot compute and store all patch embeddings in memory at the same time. We instead propose to iterate over patches, $I$ at a time, and autoregressively maintain a set of top $M$ patch embeddings. In other words, say $P_M^t$ is a buffer of $M$ patch embeddings at time step $t$ and $P_I^{t+1}$ are the next $I$ patch embeddings in the autoregressive update step. We run the following for $T$ iterations:

$$P_M^{t+1} = \text{Top-M}\{P_M^t \cup P_I^{t+1} \mid \boldsymbol{a}^{t+1}\}, \tag{1}$$

where $T = \lceil (N-M)/I \rceil$, $P_M^0 = \{\boldsymbol{X}_1, \ldots, \boldsymbol{X}_M\}$ is the initial buffer of embeddings $\boldsymbol{X}_{\{1,\ldots,M\}}$ and $\boldsymbol{a}^{t+1} \in \mathbb{R}^{M+I}$ are attention scores of considered patches at iteration $t+1$, based on which the selection in Top-M is made. These attention scores are obtained from the cross-attention transformer as described in Sect. 2.2. The output of IPS is a set of $M$ patches corresponding to $P_M^T$. Note that both patch embedding and patch selection are executed in no-gradient and evaluation mode. The former entails that no gradients are computed and stored, which renders IPS runtime and memory-efficient. The latter ensures deterministic patch selection behavior when using BatchNorm and Dropout.

**Data loading** We introduce three data loading strategies that trade off memory and runtime efficiency during IPS. In **eager loading**, a batch of images is loaded onto the GPU and IPS is applied to each image in parallel—this is the fastest variant but requires storing multiple images at once. In **eager sequential loading**, individual images are loaded onto the GPU and thus patches are selected for one image at a time until a batch of $M$ patches per image is selected for training. This enables the processing of different sequence lengths without padding and reduces memory usage at the cost of a higher runtime. In contrast, **lazy loading** loads a batch of images onto CPU memory. Then, only patches and corresponding embeddings pertinent to the current iteration are stored on the GPU—this decouples GPU memory usage from the image size, again at the cost of a higher runtime.

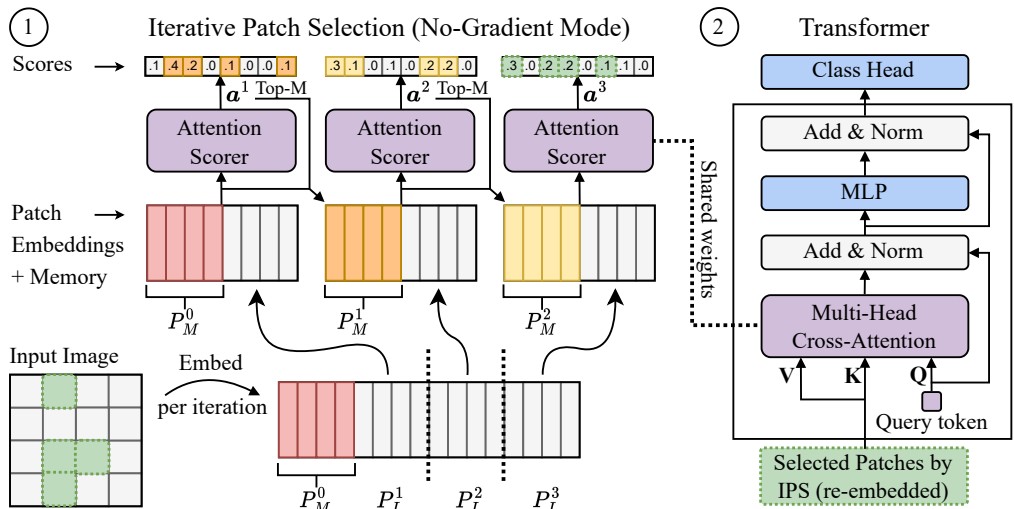

Figure 2: Left: Illustration of Iterative Patch Selection running for 3 iterations with $N = 16$, $M = 4$ and $I = 4$. Right: Cross-attention transformer module that aggregates patches selected by IPS.

## 2.2 TRANSFORMER-BASED PATCH AGGREGATION

After selecting $M$ patches in the IPS stage, these patches are embedded again in gradient and training mode, and the patch aggregation module can aggregate the resulting embeddings $\boldsymbol{X}^* \in \mathbb{R}^{M \times D}$ using a convex combination:

$$\boldsymbol{z} = \sum_{m=1}^{M} a_m \boldsymbol{X}_m^*. \tag{2}$$

Each attention score $a_m$ is the result of a function of the corresponding patch embedding $\boldsymbol{X}_m^*$: $a_{\{1,...,M\}} = \text{Softmax}\left(f^\theta(\boldsymbol{X}_1^*), ..., f^\theta(\boldsymbol{X}_M^*)\right)$ and can be learned by a neural network $f^\theta$ parameterized by $\theta$. We observe that the weighted average in Eq. 2 constitutes a cross-attention layer by defining $f^\theta$ as follows:

$$f^\theta\left(\boldsymbol{X}_m^*\right) = \frac{\boldsymbol{Q}\boldsymbol{K}^T}{\sqrt{D_k}}, \tag{3}$$

where queries $\boldsymbol{Q} = \boldsymbol{q}^\phi \boldsymbol{W}^q$ and keys $\boldsymbol{K} = \boldsymbol{X}_m^* \boldsymbol{W}^k$ are linear transformations of a learnable query token $\boldsymbol{q}^\phi \in \mathbb{R}^D$ and embedding $\boldsymbol{X}_m^*$, respectively, with weights $\boldsymbol{W}^q, \boldsymbol{W}^k \in \mathbb{R}^{D \times D_k}$ and $D_k$ being the projected dimension. Furthermore, instead of aggregating patch embeddings directly, we aggregate a linear projection called values in typical transformer notation: $\boldsymbol{V} = \boldsymbol{X}_m^* \boldsymbol{W}^v$, with $\boldsymbol{W}^v \in \mathbb{R}^{D \times D_v}$ and $D_v$ being the value dimension. In practice, we use multi-head cross-attention and place it in a standard transformer (Vaswani et al., 2017), optionally adding a $1D-$sinusoidal position encoding (details in Appendix B). Note that there is a close connection between the patch selection and patch aggregation modules. They both share the cross-attention layer. Whereas IPS runs in no-gradient mode and cannot train its parameters, by sharing them with the patch aggregation module, one can still learn to select patches relevant for the downstream task.

## 3 RELATED WORK

The setting described in Sect. 2 constitutes a MIL problem (Maron & Lozano-Pérez, 1997), where an image is seen as a bag that contains multiple patch-level features called instances. The goal is then to predict the label of unseen bags given their instances. However, training labels are only available for bags, not for instances. The model must thus learn to attend to patches that best represent the image for a given task. Eq. 2 provides such a mechanism and is known as MIL pooling function. In particular, our cross-attention layer follows the weighted collective assumption stating that all instances contribute to the bag representation, albeit to varying degrees according to the attention scores (Foulds & Frank, 2010; Amores, 2013). A MIL approach similar to ours is DeepMIL (Ilse et al., 2018), which uses a gated attention mechanism to compute scores for the aggregation of $N$

patches: $f^\theta(\boldsymbol{X}_n) = \boldsymbol{w}(\tanh(\boldsymbol{X}_n\boldsymbol{U}) \odot \mathrm{sigm}(\boldsymbol{X}_n\boldsymbol{V}))^T$, with $\boldsymbol{U}, \boldsymbol{V} \in \mathbb{R}^{D \times D_h}$ and $\boldsymbol{w} \in \mathbb{R}^{D_h}$ being parameters, and $D_h$ hidden nodes. In contrast, our multi-head cross-attention layer provides more capacity by computing multiple intermediate bag representations. Furthermore, it is a MIL pooling function that can be naturally integrated into a standard and field-proven transformer module.

Other MIL methods common for binary classification of large images classify each patch and use max-pooling or a variant thereof on the class scores for patch selection and training (Campanella et al., 2018; Zhou et al., 2018; Lerousseau et al., 2020). Our method is more general and can be used for multi-{task, label, class} classification while being able to attend to multiple patches. In early experiments, we found that max-pooling in the baseline leads to overfitting. Instead, we employ TopMIL (based on Campanella et al. (2019)) as a baseline, which selects the top $M$ patches in a single iteration according to the maximum class logit of each patch. Selected patches are then re-embedded, classified, class logits are averaged, and an activation function (e.g., softmax) is applied.

Another line of work seeks to reduce memory usage for the processing of high-resolution images by first identifying regions of interest in low resolution and then sampling patches from these regions in high resolution (Katharopoulos & Fleuret, 2019; Cordonnier et al., 2021). This is also related to early attention approaches that rapidly process the entire image based on simple features (Culhane & Tsotsos, 1992; Koch & Ullman, 1987), as well as to human attention where saccadic eye movements may be informed by peripheral vision (Burt, 1988). However, especially with gigapixel images, low-resolution images may either still be too large to encode, or too small to identify informative regions. In contrast, IPS extracts deep features from high-resolution patches to estimate attention scores.

For very large images, patch-wise self-supervised learning is a promising way to reduce memory usage (Dehaene et al., 2020; Chen et al., 2022; Li et al., 2021). After pre-training, low-dimensional features are extracted and used to train a downstream network. Although the input size decreases considerably in this way, it may still require too much memory for very long sequences. Pre-training can be used orthogonally to IPS to process sequences of any length, as demonstrated in Sect. 4.2.

Our method is related to Sparse Mixture of Experts, a conditional computation approach in which specialized subnetworks (experts) are responsible for the processing of specific input patterns (Jacobs et al., 1991). Specifically, Shazeer et al. (2017) and Riquelme et al. (2021) use a gating function to decide which expert is assigned to which part of the input. This gating function is non-differentiable but shares its output with a differentiable aggregation operation that combines the expert's outputs. On a high level this is similar to how we share the cross-attention module between no-gradient mode IPS and the gradient mode patch aggregator. Attention Sampling (Katharopoulos & Fleuret, 2019) also reuses aggregation weights for patch sampling. IPS uses Top-M instead.

## 4 EXPERIMENTS

We evaluate the performance and efficiency of our method on three challenging datasets from a variety of domains and training regimes: Multi-class recognition of distant traffic signs in megapixel images, weakly-supervised classification in gigapixel whole-slide images (WSI) using self-supervised representations, and multi-task learning of inter-patch relations on a synthetic megapixel MNIST benchmark. All training and baseline hyperparameters are provided in Appendix D.

**Baselines**  To show that IPS selects salient patches, two other patch selection methods, Random Patch Selection (RPS) and Differential Patch Selection (DPS) (Cordonnier et al., 2021), are used for comparison, both employing the same transformer as IPS. Next, to verify that the cross-attention transformer is a strong MIL pooling operator, it is compared to DeepMIL (Ilse et al., 2018) and TopMIL as introduced in Sect. 3. Finally, we compare to a standard CNN applied on the original image and a smaller sized version to assess how resolution affects performance.

**Metrics**  Each of the main experiments is performed five times with random parameter initializations, and the average classification performance and standard deviation are reported on the test set using either accuracy or AUC score. We also report computational efficiency metrics, including maximum GPU memory usage (VRAM) and training runtime for a batch size of 16 and various values of $M$. The runtime is calculated for a single forward and backward pass and is averaged over all iterations of one training epoch excluding the first and last iterations. Both metrics are calculated on a single NVIDIA A100 GPU in all experiments.

## 4.1 Traffic Signs Recognition

We first evaluate our method on the Swedish traffic signs dataset, which consists of 747 training and 684 test images with 1.3 megapixel resolution, as in Katharopoulos & Fleuret (2019). Each image shows a speed limit sign of 50, 70, 80 km/h or no sign. This problem requires high-resolution images to distinguish visually similar traffic signs, some of which appear very small due to their distance from the camera (Fig. 5 left). We resize each image to $1200 \times 1600$ and extract 192 non-overlapping patches of size $100 \times 100$. Due to the small number of data points, a ResNet-18 with ImageNet-1k weights is used as encoder for all methods and then fine-tuned. For IPS, we use eager loading and set $I = 32$.

Table 1: Traffic sign recognition: Our IPS transformer outperforms all baselines using 1.2 GB of VRAM while being faster than the CNN. DPS uses $N = 768$ patches (see Appendix D.5). VRAM is reported in gigabytes and time in milliseconds.

|  | $M$ | Acc. [%] | VRAM | Time |
|---|---|---|---|---|
| IPS Transformer | 192 | 98.4 $\pm 0.5$ | 14.2 | 323 |
| (ours) | 10 | 98.6 $\pm 0.5$ | 1.3 | 126 |
|  | 2 | 98.5 $\pm 0.8$ | 1.2 | 114 |
| RPS Transformer | 10 | 50.7 $\pm 2.6$ | 1.3 | 31 |
|  | 2 | 28.3 $\pm 3.2$ | 0.9 | 26 |
| DPS Transformer | 10 | 94.0 $\pm 2.2$ | 3.5 | 337 |
|  | 2 | 97.2 $\pm 0.7$ | 2.5 | 296 |
| TopMIL | 10 | 97.7 $\pm 0.5$ | 4.5 | 109 |
| TopMIL | 2 | 98.2 $\pm 0.5$ | 4.5 | 96 |
| DeepMIL | 192 | 96.2 $\pm 1.6$ | 14.2 | 323 |
| DeepMIL+ | 192 | 97.7 $\pm 1.3$ | 14.3 | 323 |
| CNN | – | 84.1 $\pm 4.7$ | 13.4 | 231 |
| CNN 0.3× | – | 76.3 $\pm 0.9$ | 1.4 | 83 |

Table 1 shows that the average accuracy of the IPS transformer is consistently high regardless of the value of $M$. For $M \in \{2, 10\}$, memory usage is more than $10\times$ less than for the CNN, and for $M = 2$, training time is about half that of the CNN. As expected, RPS does not select salient patches due to the low signal-to-noise ratio in the data. DPS localizes the salient patches but performs worse than IPS in all metrics and is sensitive to the choice of $M$. The MIL baselines perform well in general, and TopMIL comes closest to IPS in accuracy. However, TopMIL requires knowledge about the number of informative patches, since the logits are averaged—already for $M = 10$, the performance is slightly degraded. In addition, TopMIL is faster but less memory-efficient than IPS because in TopMIL all patches are scored at once. DeepMIL, on the other hand, lacks a patch selection mechanism and hence includes all patches in the aggregation, which increases training time and VRAM. For $M = 192$, DeepMIL differs from the IPS transformer only in the aggregation function, yet the transformer performs 2.2 percentage points better on average. We also report results for DeepMIL+, which has a similar capacity to the transformer for a fairer comparison. It achieves an average accuracy of 97.7%, less than 98.4% of IPS. A CNN that takes the entire image as input and has no attention mechanism performs much worse and is inefficient. Downscaling the image by a factor of 3 further degrades the CNN's performance, as details of distant traffic signs are lost.

For a fair comparison against prior work, we used the same transformer and pre-trained model where applicable. This results in better numbers than reported by DPS, which achieves 97.2% in Table 1 vs. 91.7% in Cordonnier et al. (2021). Nonetheless, our IPS transformer improves on that by up to 1.4 percentage points while being more memory and runtime efficient.

## 4.2 Gigapixel WSI Classification

Next, we consider the CAMELYON16 dataset (Litjens et al., 2018), which consists of 270 training and 129 test WSIs of gigapixel resolution for the recognition of metastases in lymph node cells. The task is formulated as a weakly-supervised binary classification problem, i.e., only image-level labels about the presence of metastases are used. The WSIs come with multiple resolutions, but we attempt to solve this task with the highest available resolution ($40\times$ magnification). Since a large portion of the images may not contain any cells, Otsu's method (Otsu, 1979) is first applied to filter out the background. Only patches with at least 1% of the area being foreground are included—a sensitive criterion that ensures that no abnormal cells are left out. Then, non-overlapping patches of size $256 \times 256$ are extracted. This gives a total of 28.6 million patches (avg.: 71.8k, std.: 37.9k), with the largest WSI having 251.8k patches (about 20% of the size of ImageNet-1k).

Table 2: CAMELYON16: IPS achieves performance close to the strongly-supervised state-of-the-art. †: Average number of patches is reported for these methods, NR: $N$ and $M$ are unknown.

|  | $M$ | Total ($N$) | Pretr. | AUC [%] | VRAM [GB] | Time [ms] |
|---|---|---|---|---|---|---|
| IPS Transformer (ours) | $5k$ | all | BYOL | $98.1_{\pm0.3}$ | 4.7 | 355 |
|  | $1k$ | all | BYOL | $97.5_{\pm0.2}$ | 4.1 | 313 |
|  | 100 | all | BYOL | $97.3_{\pm0.6}$ | 4.0 | 313 |
|  | $5k$ | $70k$ | BYOL | $95.1_{\pm1.7}$ | 2.9 | 286 |
| DeepMIL | $10k$ | $10k$ | BYOL | $84.1_{\pm1.9}$ | 4.5 | 26 |
|  | $50k$ | $50k$ | BYOL | $93.8_{\pm1.4}$ | 22.5 | 110 |
|  | $70k$ | $70k$ | BYOL | $94.5_{\pm1.1}$ | 31.5 | 150 |
| TopMIL | $5k$ | $70k$ | BYOL | $71.8_{\pm2.0}$ | 19.8 | 84 |
|  | $1k$ | $70k$ | BYOL | $76.2_{\pm2.9}$ | 19.8 | 76 |
|  | 100 | $70k$ | BYOL | $84.4_{\pm5.9}$ | 19.8 | 74 |
| DSMIL-LC (Li et al., 2021) | $8k^{\dagger}$ | $8k^{\dagger}$ | SimCLR | 91.7 | – | – |
| CLAM (Lu et al., 2021) | $42k^{\dagger}$ | $42k^{\dagger}$ | CPC | 93.6 | – | – |
| CLAM-SB (Wang et al., 2022) | NR | NR | SRCL | 94.2 | – | – |
| DeepMIL (Dehaene et al., 2020) | $10k$ | $10k$ | MoCo v2 | 98.7 | – | – |
| Challenge Winner (Wang et al., 2016) | – | – | None | 92.5 | – | – |
| Strong superv. (Bejnordi et al., 2017) | – | – | None | 99.4 | – | – |

Due to the large volume of patches, it is impractical to learn directly from pixels. Instead, we train BYOL (Grill et al., 2020), a state-of-the-art self-supervised learning algorithm, with a ResNet-50 encoder and extract features for all patches and WSIs (see Appendix D.6 for details). Each patch is thus represented as a 2,048-dimensional vector, which is further projected down to 512 features and finally processed by the respective aggregator. Note that even when processing patches at the low-dimensional feature level, aggregating thousands of patch features can become prohibitively expensive. For example, with DeepMIL, we can aggregate around 70k patch features before running out of memory on an A100 at a batch size of 16. Therefore, in each of the 5 runs, a random sample of up to 70k patch features is drawn for the baseline MIL methods, similar to Dehaene et al. (2020). In contrast, IPS can easily process all patches. Due to the varying number of patches in each WSI, a batch cannot be built without expensive padding. We thus switch to eager sequential loading in IPS, i.e., slides are loaded sequentially and $M$ patch features per image are selected and cached for subsequent mini-batch training. In all IPS experiments, $I = 5k$.

We report AUC scores, as in the challenge (Bejnordi et al., 2017), in Table 2. Best performance is achieved by IPS using $M = 5k$, with an improvement of more than 3 percentage points over the runner-up baseline. We also outperform the CAMELYON16 challenge winner (Wang et al., 2016), who leveraged patch-level labels, and approach the result of the strongly-supervised state of the art (Bejnordi et al., 2017). The memory usage in IPS is 4–5 GB depending on $M$. However, IPS has a higher runtime because all patches are taken into account and eager sequential loading is used. For comparison, we also reduce the total number of patches before the IPS stage to 70k (using the same random seeds) and observe slightly better performance than DeepMIL. DPS was not considered since the downscaled image exceeded our hardware limits. We also tested CNNs trained on downscaled images up to 4k×4k pixels but they perform worse than a naive classifier.

Several recent works adopt a pre-trained feature extractor and report results for CAMELYON16 (see bottom of Table 2). Given their respective setups, most consider all patches for aggregation but make compromises that one can avoid with IPS: A batch size of 1 (Wang et al., 2022; Li et al., 2021), training with features from the third ResNet-50 block, which effectively halves the size of the embeddings (Lu et al., 2021), or using smaller resolutions (Dehaene et al., 2020; Li et al., 2021). The work by Dehaene et al. (2020) achieves a slightly better result than our IPS transformer (avg. AUC: 98.1% vs. 98.7%) by using DeepMIL on a random subset of up to $10k$ patches. We do not assume that this is due to our transformer, as the DeepMIL baseline reproduces their aggregation function yet performs significantly worse. Instead, it could be due to a different pre-training algorithm (MoCo v2), pre-processing (closed-source U-Net to filter out the background), or lower magnification level (20×), which reduces the total number of patches (avg. 9.8k vs. 71.8k in ours).

Table 3: Results for megapixel MNIST. The IPS Transformer accurately solves all relational tasks with a minimal number of patches and can leverage overlapping patches with small memory usage.

| | $M$ | Total ($N$) | Majority | Max | Top | Multi-label | VRAM [GB] | Time [ms] |
|---|---|---|---|---|---|---|---|---|
| IPS Transformer | 900 | 3,481 | 99.8 $_{\pm 0.1}$ | 99.0 $_{\pm 0.2}$ | 95.0 $_{\pm 1.1}$ | 93.6 $_{\pm 3.3}$ | 15.5 GB | 570 ms |
| (ours) | 100 | 3,481 | 99.7 $_{\pm 0.1}$ | 98.8 $_{\pm 0.2}$ | 95.9 $_{\pm 0.8}$ | 96.1 $_{\pm 0.7}$ | 2.2 GB | 326 ms |
| | 5 | 3,481 | 97.5 $_{\pm 1.0}$ | 98.4 $_{\pm 0.1}$ | 96.7 $_{\pm 0.5}$ | 90.2 $_{\pm 1.0}$ | 1.1 GB | 301 ms |
| | 900 | 900 | 99.1 $_{\pm 0.2}$ | 93.7 $_{\pm 0.5}$ | 91.7 $_{\pm 1.4}$ | 82.5 $_{\pm 3.0}$ | 15.0 GB | 314 ms |
| | 100 | 900 | 98.9 $_{\pm 0.3}$ | 94.1 $_{\pm 0.6}$ | 92.3 $_{\pm 0.7}$ | 84.2 $_{\pm 2.0}$ | 1.8 GB | 114 ms |
| | 5 | 900 | 94.8 $_{\pm 0.7}$ | 92.4 $_{\pm 0.8}$ | 91.0 $_{\pm 1.0}$ | 72.1 $_{\pm 3.0}$ | 0.7 GB | 89 ms |
| RPS Transformer | 900 | 3,481 | 82.5 $_{\pm 1.6}$ | 78.5 $_{\pm 1.9}$ | 77.6 $_{\pm 1.0}$ | 47.0 $_{\pm 2.4}$ | 15.4 GB | 314 ms |
| | 100 | 3,481 | 33.8 $_{\pm 1.3}$ | 34.9 $_{\pm 1.3}$ | 29.1 $_{\pm 1.5}$ | 0.9 $_{\pm 0.3}$ | 2.2 GB | 47 ms |
| DPS Transformer | 100 | 900 | 97.2 $_{\pm 1.0}$ | 93.2 $_{\pm 0.4}$ | 92.3 $_{\pm 0.6}$ | 81.4 $_{\pm 1.5}$ | 14.0 GB | 353 ms |
| | 5 | 900 | 88.1 $_{\pm 3.1}$ | 81.7 $_{\pm 2.6}$ | 80.1 $_{\pm 3.6}$ | 48.3 $_{\pm 5.7}$ | 2.2 GB | 285 ms |
| TopMIL | 900 | 900 | 98.2 $_{\pm 0.4}$ | 90.0 $_{\pm 1.0}$ | 52.9 $_{\pm 0.6}$ | 71.9 $_{\pm 4.0}$ | 15.2 GB | 381 ms |
| | 100 | 900 | 98.4 $_{\pm 0.3}$ | 92.4 $_{\pm 0.3}$ | 53.8 $_{\pm 1.0}$ | 78.0 $_{\pm 3.0}$ | 4.6 GB | 105 ms |
| | 5 | 900 | 97.2 $_{\pm 0.6}$ | 91.4 $_{\pm 0.6}$ | 71.2 $_{\pm 1.0}$ | 72.3 $_{\pm 1.7}$ | 4.6 GB | 73 ms |
| DeepMIL | 900 | 900 | 98.8 $_{\pm 0.2}$ | 93.8 $_{\pm 0.2}$ | 92.3 $_{\pm 0.9}$ | 80.7 $_{\pm 1.2}$ | 15.0 GB | 316 ms |
| CNN | – | – | 99.3 $_{\pm 0.4}$ | 91.9 $_{\pm 0.6}$ | 57.0 $_{\pm 1.1}$ | 71.7 $_{\pm 9.0}$ | 13.7 GB | 209 ms |
| CNN 0.3x | – | – | 90.8 $_{\pm 0.7}$ | 76.7 $_{\pm 1.3}$ | 55.8 $_{\pm 0.4}$ | 42.3 $_{\pm 2.6}$ | 1.6 GB | 30 ms |

## 4.3 INTER-PATCH REASONING

A single informative patch was sufficient to solve the previous tasks. In contrast, megapixel MNIST introduced in Katharopoulos & Fleuret (2019) requires the recognition of multiple patches and their relations. The dataset consists of 5,000 training and 1,000 test images of size 1,500×1,500. We extract patches of size 50×50 without overlap ($N = 900$) or with 50% overlap ($N = 3,481$). In each image, 5 MNIST digits are placed, 3 of which belong to the same class and 2 of which to other classes. In addition, 50 noisy lines are added (see Fig. 5 right). The task is then to predict the *majority* class. We found that this problem can be solved well by most baselines, so we extend the setup with three more complex tasks: *max*: detect the maximum digit, *top*: identify the topmost digit, *multi-label*: the presence/absence of all classes needs to be recognized. We frame this as a multi-task learning problem and use 4 learnable query tokens in the cross-attention layer, i.e. 4 task representations are learned. Similarly, 4 gated-attention layers are used for DeepMIL. All methods also utilize multiple classification heads and add a $1D-$sinusoidal position encoding to the patch features. For the CNNs, a sinusoidal channel is concatenated with the input. A simplified ResNet-18 consisting of 2 residual blocks is used as the encoder for all methods. For IPS, we use eager loading and set $I = 100$.

The results in Table 3 show that almost all methods were able to obtain high accuracies for the tasks *majority* and *max*. However, only DeepMIL, DPS and IPS were able to make appropriate use of the positional encoding to solve task *top*. For the hardest task, *multi-label*, only the IPS transformer obtained accuracies above 90% by using overlapping patches. However, for both TopMIL and Deep-MIL, we were not able to use overlapping patches due to out-of-memory errors (on a single A100). Interestingly, IPS achieves high performance throughout with $M = 100$ using overlapping patches at a memory usage of only 2.2 GB while maintaining a competitive runtime. Furthermore, the IPS transformer also achieves high performance with $M = 5$ patches, which is the minimum number of patches required to solve all tasks.

## 4.4 ABLATION STUDY

**Effect of image size** How does input size affect performance and efficiency in IPS? To address this question, we generate new megapixel MNIST datasets by scaling the image size from 1k to 10k pixels per side (400 to 40k patches) and attempt to solve the same tasks as in the main experiment. We also linearly scale the number of noise patterns from 33 to 333 to make the task increasingly difficult. Fig. 3 shows a comparison of memory usage, runtime and accuracy (task *majority*) for different data loading options of our IPS transformer ($M = 100$, $I = 100$, non-overlapping patches, default batch size of 16), as well as the baseline CNN.

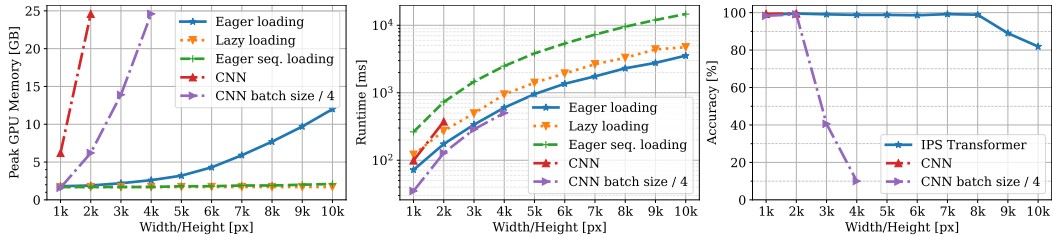

Figure 3: Peak GPU memory (left), training runtime (center) and accuracy (right) as a function of image size in megapixel MNIST: With IPS and lazy loading, memory usage can be kept constant. High performance is mostly maintained compared to the baseline CNN.

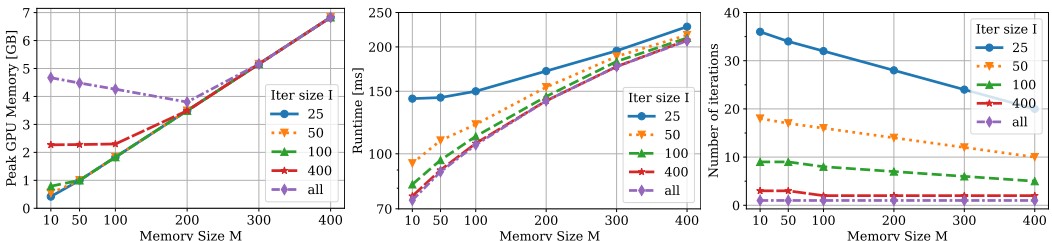

Figure 4: VRAM, training runtime and number of iterations for different combinations of $M$ and $I$.

The CNN exhibits a high memory footprint that quickly exceeds the hardware limits even when using a reduced batch size of 4. In IPS with eager loading, the VRAM is dominated by the batch of inputs and can be reduced by sequential loading. In contrast, with lazy loading the memory usage is kept constant at 1.7 GB regardless of the input size. The runtime in eager loading is less than that of the CNN and almost approaches a CNN using a batch size of 4. Lazy loading is faster than eager sequential loading, suggesting that the latter should only be used in situations where different sequence lengths need to be processed. In terms of performance, we note that the IPS transformer achieves high accuracy for up to 8k pixels and decreases only slightly starting from 9k pixels likely due to the declining signal-to-noise ratio (at 10k pixels, only 5 out of 40k patches are relevant). In comparison, the CNN performance drops rapidly starting from 3k pixels towards a naive classifier.

**Effect of $M$ and $I$** Fig. 4 shows that VRAM and runtime tend to increase with $M$ as more patches need to be processed in training mode. With increasing values of $I$, a higher VRAM but lower runtime can be observed, since IPS runs for fewer iterations. However, there are more subtle cases. Along the diagonal in Fig. 4 (left), peak VRAM is observed in the non-IPS part and thus remains constant for different values of $I$. For example, for $M = 50$, VRAM remains constant for $I \in \{25, 50, 100\}$, so setting $I = 100$ results in faster runtime without compromising memory usage. In contrast, in the upper left triangle of the graph, the VRAM is dominated by IPS in no-gradient mode. For example, $I = all$ uses $\approx$5GB of VRAM for $M = 10$ (i.e., $I = 890$). Interestingly, when processing all remaining patches in a single iteration ($I = N - M$), VRAM can be reduced by increasing $M$ up to 200, because $I$ is reduced by $M$.

**Effect of patch size** In Table 4, we investigate the effect of patch size (25–400 px per side) on the performance and efficiency of traffic sign recognition. One can notice that the accuracy is similar for all patch sizes and only slightly decreases for very small patch sizes of 25 px. As the patch size increases, the runtime for IPS tends to decrease since fewer iterations are run, while the time to embed and aggregate selected patches (non-IPS time) increases because more pixels need to be processed in training mode.

### 4.5 INTERPRETABILITY

In Fig. 5, we superimpose attention scores over images, which shows that salient patches are attended to while the background is neglected. In low signal-to-noise settings, such visualizations

Table 4: Effect of patch size on traffic sign recognition and efficiency ($M = 2$, $I = 32$, single run).

| Size [px] | Acc. [%] | VRAM [GB] | IPS Time [ms] | Non-IPS Time [ms] | Total Time [ms] |
|---|---|---|---|---|---|
| 25 | 96.7 | 0.9 | 417 | 6 | 423 |
| 50 | 98.7 | 1.0 | 167 | 7 | 174 |
| 100 | 98.8 | 1.2 | 104 | 10 | 114 |
| 200 | 99.1 | 3.3 | 74 | 15 | 89 |
| 400 | 98.4 | 4.2 | 88 | 49 | 137 |

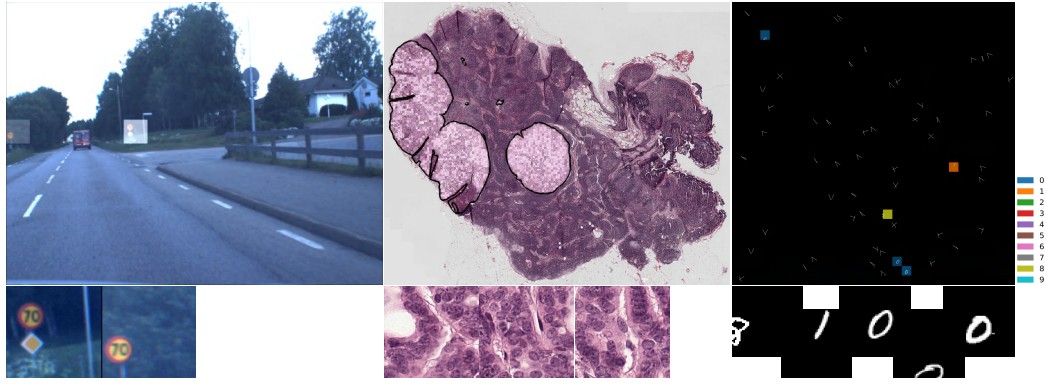

Figure 5: Patch scores and crops of most salient patches. Left: Traffic signs obtain high scores. Center: Most cancer cells (black contour) are attended. Right: Patch-level class mappings in MNIST.

can support humans to localize tiny regions of interest that might otherwise be missed (e.g., Fig. 1). However, attention scores alone may not be satisfactory when multiple objects of different classes are present. For example, attention scores in megapixel MNIST indicate relevant patches, but not to which classes these patches belong. We propose a simple fix to obtain patch-level class scores for $C$ classes. We add a new classification head: $\tilde{\boldsymbol{Y}}_m = g^\psi(\boldsymbol{X}_m^*)$, where $g^\psi : \mathbb{R}^D \to \mathbb{R}^C$ with parameters $\psi$. For training, these scores compute an image-level prediction: $\hat{\boldsymbol{y}} = \sigma(\sum_{m=1}^M a_m \tilde{\boldsymbol{Y}}_m)$, where $\sigma$ is an activation function. Crucially, we stop gradients from flowing back through $\boldsymbol{X}_m^*$ and $a_m$ such that $g^\psi$ is trained as a separate read-out head jointly with the base model. We demonstrate this for megapixel MNIST ($N = 900$), where the patch-level classifier is learned from the multi-label task, using only image-level labels. In Fig. 5 (right), we visualize the inferred per-patch class label, $\text{argmax}_c \tilde{\boldsymbol{Y}}_m$, using colors and $a_m$ using the alpha channel.

## 5 CONCLUSION

Reflecting on the experiments, one can observe that the IPS transformer performs well in various domains and training regimes and exhibits interesting capabilities: constant memory usage, learning with arbitrarily long sequences of varying lengths, multi-task learning, and modeling of inter-patch relations. However, there are also limitations. For example, while the GPU memory can be kept constant, runtime cannot. In section 4.4, we gave some intuition about what influence hyperparameters $M$ and $I$ have in this regard. Furthermore, performance can degrade if the signal-to-noise ratio becomes too small, as shown in our MNIST scaling experiment—in practice, this might be mitigated by pre-training. Nonetheless, we believe that our main contribution, maintaining a low memory footprint for high-resolution image processing, can foster interesting applications that would otherwise be prohibitively expensive. Examples of such applications are learning with multiple medical scans for diagnosis/prognosis, the processing of satellite images, prediction tasks in videos, graphs, point clouds or whole-genome sequences. Technically, it would be interesting to extend IPS to dense prediction tasks, such as segmentation or depth estimation.

## ETHICS STATEMENT

**Datasets**   CAMELYON16 is a public benchmark dataset consisting of histopathology slides obtained from tissue of human subjects. The dataset has been approved by an ethics committee, see Litjens et al. (2018) for more information. For the Swedish traffic signs dataset, we searched the data for humans and excluded one image where a face of a pedestrian was clearly visible. Megapixel MNIST builds on the widely used MNIST dataset, which contains digits that are ethically uncritical.

**Applications**   In this paper, we exclusively conduct experiments and demonstrate potential applications for human benefit, such as early traffic signs detection, which could be useful for advanced driver assistance systems, or abnormality detection in medical images, which could assist physicians. However, more care is needed before any of these systems can be deployed in real-world applications including an assessment of biases and fairness and monitoring by medical professionals where relevant. Furthermore, the attention scores are currently **not** usable as a detection method. Beyond the showcased applications, our method could be misused, just like any other visual object detection system, but now also on gigapixel images. We do not explore such applications in our paper and do not plan to do so in future work.

**Computational efficiency**   Our method enables the training of deep neural networks on arbitrary large inputs even on low-cost consumer hardware with limited VRAM, which makes AI development more accessible. All training and fine-tuning experiments on downstream tasks can be efficiently run on a single A100 GPU, with the following total training runtimes: Traffic sign recognition: 45 min ($M = 10, I = 32$, 150 epochs), megapixel MNIST: 5h ($M = 100, I = 100, N = 3{,}481$, 150 epochs), CAMELYON16: 2h ($M = 5{,}000, I = 5{,}000, N = all$, 50 epochs).

## REPRODUCIBILITY STATEMENT

Sect. 2, the pseudocode provided in Appendix A, as well as the supplementary source code on Github enable the reproduction of our method. Our metrics and baselines lean on existing work and are described in Sect. 4. Necessary steps to reproduce our experiments are described in respective subsections of Sect. 4. We provide further details about all network architectures, hyperparameters for our method and the baselines, as well as pre-processing steps in Appendix D. The tools used to measure the training runtime and peak GPU memory are listed in Appendix D.7.

## ACKNOWLEDGMENTS

We would like to thank Joan Puigcerver for suggesting relevant related work, Romeo Sommerfeld for fruitful discussions on interpretability, Noel Danz for useful discussions about applications, as well as Jonathan Kreidler, Konrad Lux, Selena Braune and Florian Sold for providing and managing the compute infrastructure. This project has received funding from the German Federal Ministry for Economic Affairs and Climate Action (BMWK) in the project DAKI-FWS (reference number: 01MK21009E).

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

APPENDIX

## A    PSEUDOCODE

---

**Algorithm 1:** Pseudocode for IPS and Patch Aggregation

---

**for** $step = 1, 2, \ldots$ **do**
    Load image patches
    Activate no-gradient and evaluation mode
    $P_M^0 =$ Embed first patches
    **for** $t = 0$ **to** $T$ **do**
        $P_I^{t+1} =$ Embed next patches
        $\boldsymbol{a}^{t+1} =$ Compute their scores
        $P_M^{t+1} =$ Select top $M$ embeddings from $P_M^t \cup P_I^{t+1}$
    **end**
    Fetch patches according to $P_M^T$
    Activate gradient and training mode
    $\boldsymbol{z} =$ Embed and aggregate selected patches
    Classify, compute loss, optimize
**end**

---

## B    MULTI-HEAD CROSS-ATTENTION AND TRANSFORMER

In practice, we use multi-head cross-attention (MCA), which computes multiple intermediate representations $\boldsymbol{z}^h, h = 1, \ldots, H$ corresponding to $H$ heads and then aggregates them:

$$\boldsymbol{z}^c = \mathrm{concat}\left(\boldsymbol{z}^1, \cdots, \boldsymbol{z}^H\right), \tag{4}$$

$$\boldsymbol{z} = \boldsymbol{z}^c \boldsymbol{W}^o, \tag{5}$$

with $\boldsymbol{z}^h \in \mathbb{R}^{D_v}$, $\boldsymbol{z}^c \in \mathbb{R}^{(D_v \cdot H)}$, $\boldsymbol{z} \in \mathbb{R}^D$ and $\boldsymbol{W}^o \in \mathbb{R}^{(D_v \cdot H) \times D}$. Note that each representation $\boldsymbol{z}^h$ is computed with head-specific attention scores $\boldsymbol{a}^h$. For patch selection, these scores are averaged over heads to obtain a scalar score per patch. The transformer module is composed of an MCA layer, a multi-layer perceptron (MLP) and Layernorm (LN) and is formulated as:

$$\boldsymbol{z} = \mathrm{MCA}(\boldsymbol{X}^* + \boldsymbol{X}^{pos}) \tag{6}$$

$$\boldsymbol{z}' = \mathrm{LN}(\boldsymbol{z} + \boldsymbol{q}^{\phi}) \tag{7}$$

$$\boldsymbol{z}^o = \mathrm{LN}(\mathrm{MLP}(\boldsymbol{z}') + \boldsymbol{z}'), \tag{8}$$

where $\boldsymbol{X}^{pos} \in \mathbb{R}^{M \times D}$ is an optional $1D-$sinusoidal position encoding and the MLP consists of two layers with ReLU non-linearity.

## C    EXTENDED RELATED WORK

**Cross-attention**    is an essential component of our method because it links the patch selection and patch aggregation modules. Therefore, we want to shed more light on its use in the literature. Cross-attention is a concept in which the queries stem from another input than the one utilized by keys and values. It has previously been employed in various transformer-based architectures and applications. In the original transformer, it is used in an autoregressive decoder for machine translation (Vaswani et al., 2017), where keys and values are computed from the encoder output, whereas queries are computed from the already translated sequence. CrossViT applies cross-attention to efficiently fuse patch encodings corresponding to varying patch sizes (Chen et al., 2021). Cross-attention can also be applied to object detection. For example, DETR defines learnable object tokens that cross-attend to patch embeddings, where the number of tokens reflects the maximum number of objects that can be detected (Carion et al., 2020). More related to our work is Query-and-Attend (Arar et al., 2022), which involves learnable queries as a substitute for local convolutions and the more computationally expensive self-attention mechanism. Another interesting architecture is Perceiver IO, which applies

cross-attention to the encoder for dimensionality reduction and to the decoder for learning task-specific outputs (Jaegle et al., 2021). TokenLearner (Ryoo et al., 2021) also reduces the number of tokens in the encoder for image and video recognition tasks but without using a cross-attention module. In our work, cross-attention serves two purposes. First, we treat it as a MIL pooling function that is used in the rear part of the network to combine information from multiple patches. Second, we take advantage of the attention scores to select a fixed number of most salient patches, which makes memory consumption independent of input size and thus allows us to process arbitrarily large images.

**Checkpointing**  Another memory-efficient approach is the StreamingCNN (Pinckaers et al., 2020), which employs gradient checkpointing to process large images sequentially through a CNN, which is demonstrated on images up to 8k×8k pixels. Due to the use of checkpoints, multiple forward and backward passes are required. Furthermore, backprop is performed for all tiles, which further slows down the training runtime. In IPS, all patches only run once through the network in fast no-gradient mode. Only selected patches then require a second forward and backward pass.

## D  EXPERIMENT DETAILS

### D.1  TRAINING

All models are trained for 150 epochs (megapixel MNIST, traffic signs) or 50 epochs (CAME-LYON16) on the respective training sets, and results are reported after the last epoch on the test set without early stopping. The batch size is 16, and AdamW with weight decay of 0.1 is used as optimizer (Loshchilov & Hutter, 2017). After a linear warm-up period of 10 epochs, the learning rate is set to 0.0003 when finetuning pre-trained networks and 0.001 when training from scratch. The learning rate is then decayed by a factor of 1,000 over the course of training using a cosine schedule.

### D.2  ARCHITECTURE COMPONENTS

Table 5 provides a high-level overview of the various components used by the baselines and our method. Note that all methods use the same encoder for a fair comparison.

Table 5: Module pipeline of baselines and our method

| # | CNN | DPS Transformer | TopMIL | DeepMIL | IPS Transformer (ours) |
|---|---|---|---|---|---|
| 1 | Encoder | Scorer + Diff. Top-M | Patch selection | Encoder | IPS |
| 2 | Class. head | Encoder | Encoder | Gated attention | Encoder |
| 3 | Activation | Cross-attention transf. | Class. head | Class. head | Cross-attention transf. |
| 4 | | Class. head | Avg. pool | Activation | Class. head |
| 5 | | Activation | Activation | | Activation |

### D.3  ENCODERS

The encoders differ depending on the dataset and are listed in Table 6. After applying each encoder, global average pooling is used to obtain a feature vector per patch. The CNN baselines also use these encoders, but the encoders are applied to the image instead of the patches.

### D.4  TRANSFORMER HYPERPARAMETERS

The hyperparameters of the cross-attention transformer follow default values (Vaswani et al., 2017) and are listed in Table 7. We slightly deviate from these settings for very small values of $M$. In particular, for $M = 2$ in traffic signs, we set $D_{inner} = D$. For both traffic signs with $M = 2$ and megapixel MNIST with $M = 5$, we set attention dropout to 0.

Table 6: Encoder architectures and pre-training for each dataset.

| Dataset | Encoder | Pre-training | # Features |
|---|---|---|---|
| Traffic signs | ResNet-18 | ImageNet-1k | 512 |
| CAMELYON16 | ResNet-50 + projector | BYOL | 512 |
| Megapixel MNIST | ResNet-18 (2 blocks) | None | 128 |

Table 7: Transformer hyperparameters. $D_{\mathrm{inner}}$ is the hidden layer size in the feed-forward MLP of the transformer block.

| Setting | Value |
|---|---|
| Heads | 8 |
| $D$ | # Features |
| $D_k$ | $D/8$ |
| $D_v$ | $D/8$ |
| $D_{\mathrm{inner}}$ | $4D$ |
| dropout | 0.1 |
| attention dropout | 0.1 |

### D.5 BASELINE HYPERPARAMETERS

In DPS, we mostly follow the setup of Cordonnier et al. (2021). We downsample the original image by 3 to obtain a low-resolution image (MNIST: $500 \times 500$, traffic signs: $400 \times 533$). As scorer, we use the first two blocks of a ResNet-18 without pre-training for megapixel MNIST and with ImageNet-1k pre-training for traffic signs, followed by a Conv layer (kernel: 3, padding: 0) with a single output channel and max pooling (kernel: 2, stride: 2). The resulting score maps have shape $30 \times 30$ for megapixel MNIST and $24 \times 32$ for traffic signs. Thus, we consider $N = 900$ patches and $N = 768$ patches for megapixel MNIST and traffic signs, respectively. We use 500 noise samples for perturbed top-K, employ gradient $L^2$ norm clipping with a maximum cutoff value of 0.1 and set perturbation noise $\sigma = 0.05$, which linearly decays to 0 over the course of training. For megapixel MNIST and $M = 5$, $\sigma$-decay was deactivated as it resulted in degraded performance towards the end of training.

Table 8: Classification heads for each dataset for DeepMIL and TopMIL

| # | Traffic signs | Megapixel MNIST | CAMELYON16 |
|---|---|---|---|
| 1 | fc (# classes) | fc (128) + ReLU | fc (128) + ReLU |
| 2 | | fc (# classes) | fc (64) + ReLU |
| 3 | | | fc (# classes) |

For DeepMIL, $D_h = 128$ following Ilse et al. (2018), whereas $D_h = 1024$ for DeepMIL+. The classification heads of the MIL baselines differ slightly between datasets and are listed in Table 8. In traffic signs recognition, a single output layer is utilized following Ilse et al. (2018). More capacity is added for DeepMIL+: fc (2,048) + ReLU, fc (512) + ReLU, fc (# classes). For megapixel MNIST, we obtained better results by adding an additional layer, which could be due to the multi-task setup. For CAMELYON16, we followed the settings of Dehaene et al. (2020). In DPS and our method, we use the same setup for all datasets: fc (# classes). The patch-level classification head introduced in Sect. 4.5 uses the same structure as the MIL methods in megapixel MNIST (Table 8 middle column).

### D.6 PRE-PROCESSING OF CAMELYON16

Since a large fraction of a WSI may not contain tissue, we first read each image in chunks, create a histogram of pixel intensities, and apply Otsu's method to the histogram to obtain a threshold

per image. We then extract non-overlapping patches of size 256×256, and retain patches with at least 1% of the area being foreground. BYOL is then trained with a ResNet-50 for 500 epochs using the training settings of Grill et al. (2020) and the augmentations listed in Table 9, which mostly follow those used in Dehaene et al. (2020). In each pre-training epoch, 270,000 patches are sampled at random, corresponding to approximately 1,000 patches per training slide. We train on 4 A100 GPUs with a batch size of 256 using mixed precision. After pre-training, each patch is center cropped with 224 px per side and 2,048-dimensional features resulting from the ResNet-50 encoder are extracted. The features are then normalized and stored in an HDF5 file. For the downstream task, we train a projector (fc (512), BatchNorm, ReLU), and its output is passed on to the aggregation module. For the pre-processing of the megapixel MNIST and traffic sign datasets, we adapt the code of Katharopoulos & Fleuret (2019).

Table 9: BYOL augmentations. Square brackets indicate varying probabilities in each of the 2 views.

| Type | Parameters | Probability |
|---|---|---|
| Rotation | $\{0°, 90°, 180°, 270°\}$ | 1.0 |
| VerticalFlip | - | 0.5 |
| HorizontalFlip | - | 0.5 |
| ResizedCrop | scale $= (0.2, 1.0)$ | 1.0 |
| ColorJitter | brightness $= 0.8$, contrast $= 0.8$, saturation $= 0.8$, hue $= 0.2$ | 0.8 |
| Grayscale | - | 0.2 |
| GaussianBlur | kernel size $= 23$ | $[1.0, 0.1]$ |
| Solarize | threshold $= 150$ | $[0.0, 0.2]$ |

### D.7 COMPUTATIONAL EFFICIENCY

We measure the training runtime for a full forward and backward pass without data loading for a batch size of 16, and report the average of all but the first and last iterations of one epoch. For time recording, we utilize `torch.cuda.Event`, which can be used as follows to measure the elapsed time for a single iteration:

```python
import torch
start_event = torch.cuda.Event(enable_timing=True)
end_event = torch.cuda.Event(enable_timing=True)
start_event.record()
#forward and backward pass
end_event.record()
torch.cuda.synchronize()
time = start_event.elapsed_time(end_event)
```

After running an experiment for one epoch, the following code can be used to measure the peak GPU memory usage:

```python
mem_stats = torch.cuda.memory_stats()
peak_gb = mem_stats["allocated_bytes.all.peak"] / 1024 ** 3
```

## E  IPS WITH DEEPMIL

We want to explore how IPS performs with a different aggregation module than ours. Like our cross-attention module, the gated attention mechanism used in DeepMIL (Ilse et al., 2018) also assigns attention scores to each patch. We thus train IPS jointly with DeepMIL on megapixel MNIST (multi-task) and report results for each of 5 runs in Table 10, for the *majority* task with $N = 900$, $I = 100$ and different values of $M$.

Table 10: Performance of DeepMIL when combined with IPS.

| $M$ | Accuracy per run (sorted) | | | | |
|-----|------|------|------|------|------|
| 5 | 9.1 | 9.6 | 10.0 | 10.0 | 94.3 |
| 100 | 10.0 | 98.6 | 98.7 | 98.7 | 99.2 |
| 200 | 98.5 | 98.7 | 98.9 | 99.0 | 99.1 |

For $M = 5$, one can observe that the model was able to localize the digits in only one out of five runs. However, when increasing the memory buffer to $100$, only one run failed, and for $M = 200$ all runs were successful. For IPS + transformer, all runs achieved over 90% accuracy, even for $M = 5$. The average accuracy of IPS + DeepMIL for $M = 200$ is $98.8\%$, which is on par with IPS+transformer for $M = 100$ ($98.9\%$). This shows that IPS can be used with different attention-based aggregation modules, however IPS + transformer is more robust for small values of $M$.

## F  ADDITIONAL VISUALIZATIONS

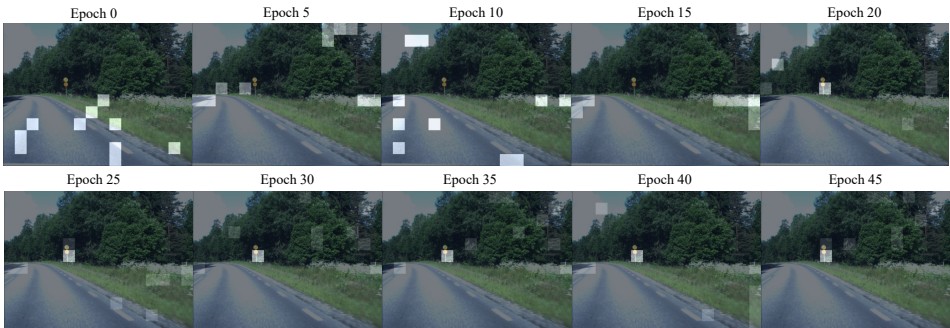

Figure 6: Evolution of attention scores in IPS for $M = 10$. The traffic sign (80 km/h) is localized starting from epoch 20. The entropy of the attention distribution appears to decrease during training.

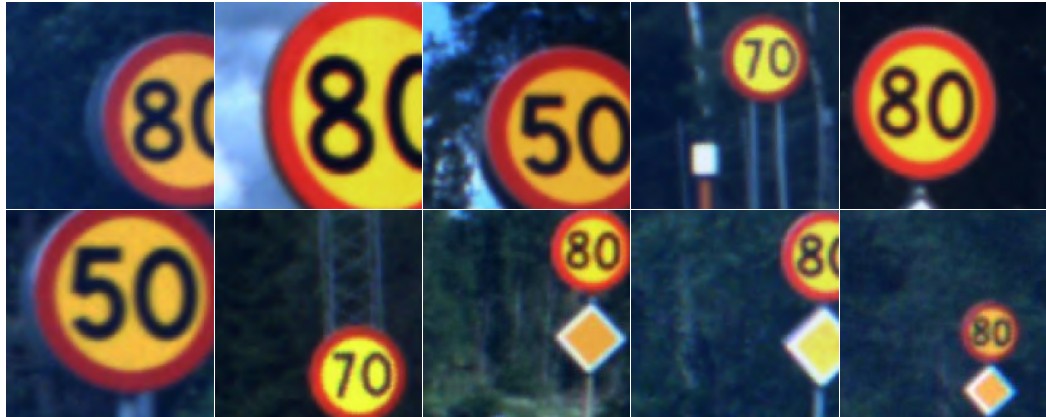

Figure 7: Patches with the highest attention scores (decreasing from left-to-right then top-to-bottom) out of all images from the traffic sign recognition test set.

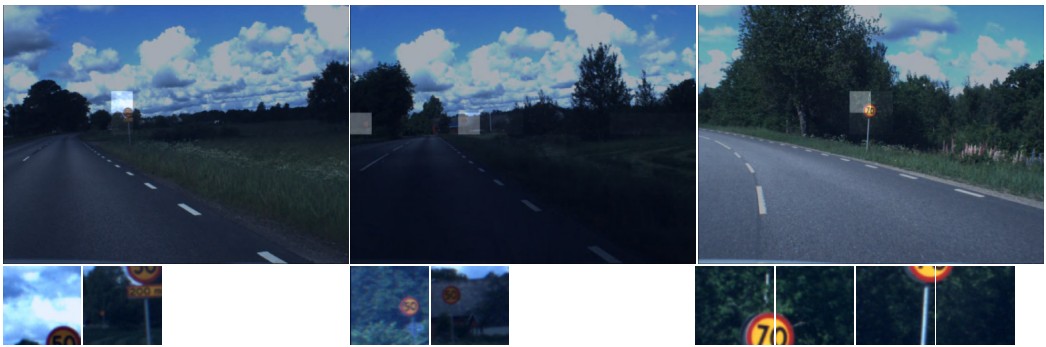

Figure 8: Attention scores and crops of the most salient patches for example test images in the traffic sign recognition dataset.

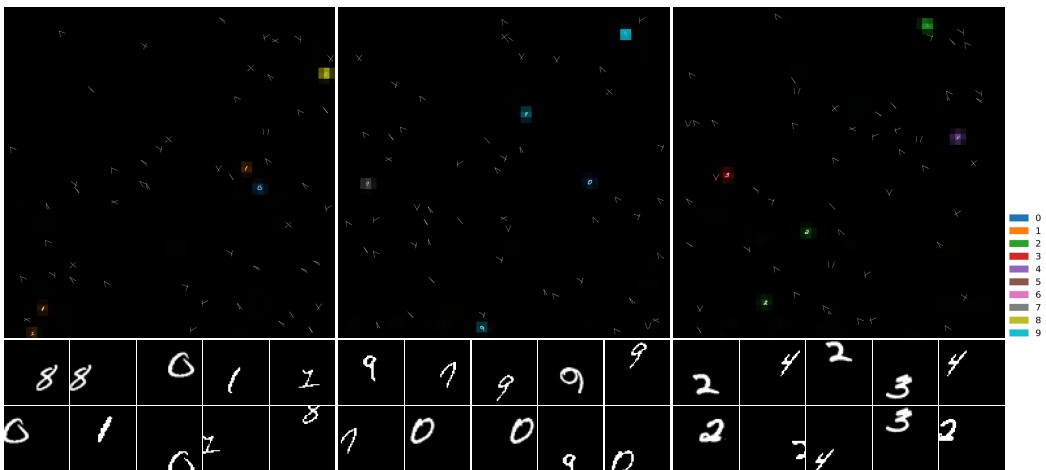

Figure 9: Patch-level classification scores and crops of patches that obtained highest attention scores (decreasing from left-to-right then top-to-bottom per image) in megapixel MNIST ($N = 3{,}481$).

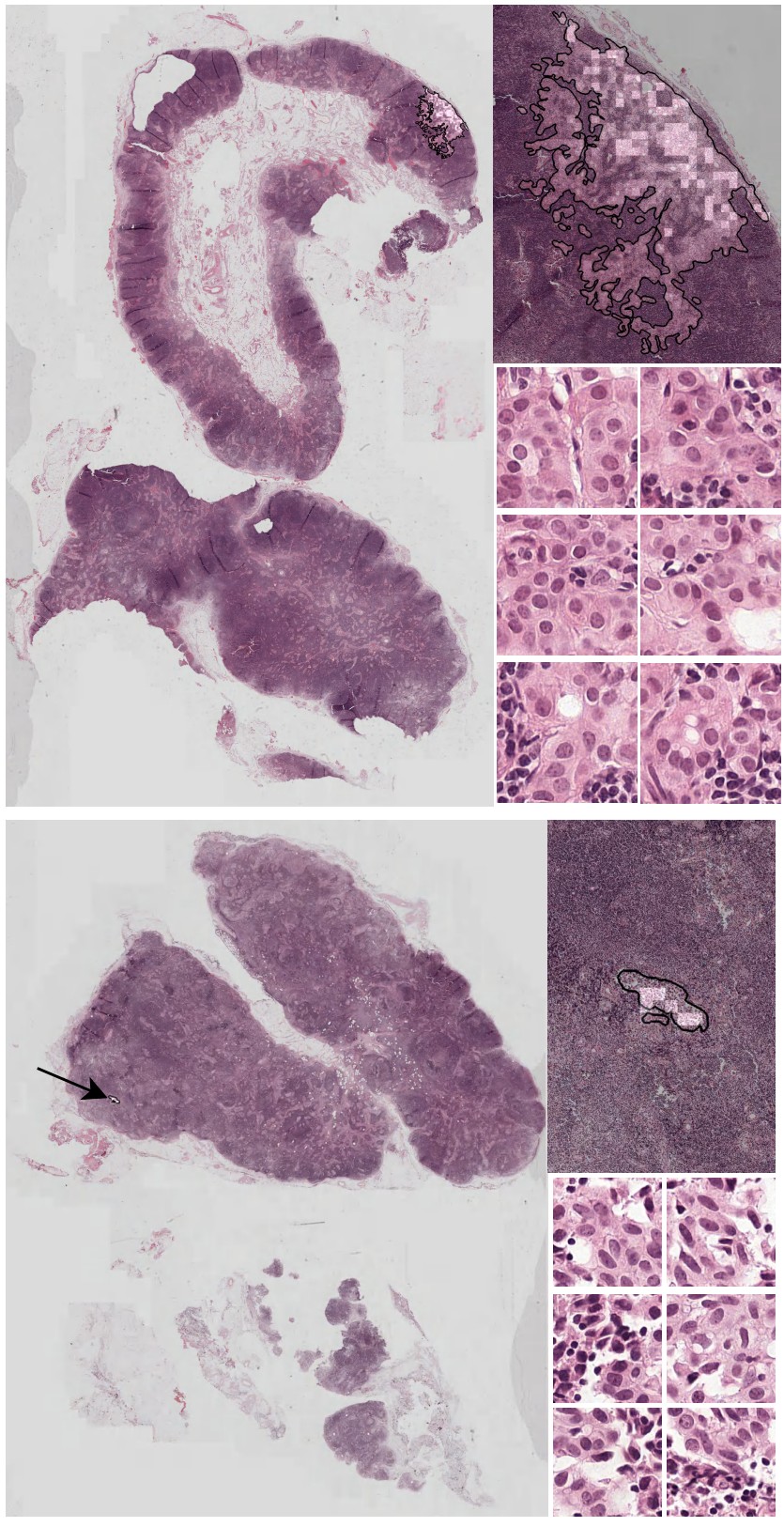

Figure 10: Attention maps overlaid on the input image and crops of the most salient patches for two examples of the CAMELYON16 dataset.

