# OpenReview forum: "Iterative Patch Selection for High-Resolution Image Recognition"
_ICLR.cc/2023/Conference — ICLR 2023 poster_

### Official Review · Reviewer_jHHu · 2022-10-21

**Confidence:** 3
**Correctness:** 3
**Technical Novelty And Significance:** 2
**Empirical Novelty And Significance:** 3
**Recommendation:** 8

**Clarity, Quality, Novelty And Reproducibility:**

Overall the paper is up to ICLR standards. There are some clarity issues, which I highlight above, that could improve the paper's readability.

**Strength And Weaknesses:**

Strengths:

S1: The paper addresses an important problem that plagues graduate students and researchers who may not have access to the latest and most expensive clusters of GPUs, namely that of being able to do vision under limited resources. It should therefore be of interest to the ICLR community.

S2: the paper is well written and is up to ICLR standards. However as I indicate below, in certain places the authors should move some of the details from the appendix into the main text.


Weaknesses:

W1: Fig 1: it is unclear how the patch scores are shown in this figure

W2: under Eq.1, X_1,...,X_M is not well defined.

W3: page 2, last paragraph: the relevant information is really in the appendix. I suggest the authors add more details here. This is currently too high level.

W4: under Eq 3, the variables just under Eq 3 need to be discussed in more detail

W5: in the paper's prior work, the authors should make a better connection to early approaches for simulating saccades in vision and attention beams. Why do humans have attention?

W6: Appendix A pseudocode: this is not pseudocode, it is actual code. I suggest edits to make it more readable.

**Summary Of The Paper:**

The authors deal with the problem of training neural networks when the input images are very large and thus cannot fit entirely into GPU memory (ie, cases where even a batch size of 1 is too large). They propose an Iterative Patch Selection algorithm which selects only the most salient patches which are then aggregated into a global representation for image recognition. A transformer is used for both patch selection and aggregation. Their experiments are able to process >16 gigapixel images on a 5GB GPU.

**Summary Of The Review:**

Overall I think this paper should be of interest to the ICLR community as it helps enable big scale deep learning under limited resources. The main weakness I see is that the authors need to do a better job positioning the paper with respect to early work on attention.

---

> ### Author Response · Authors · 2022-11-15
> **Response to Reviewer jHHu**
>
> We thank the reviewer for their thorough review, positive feedback and valuable suggestions for further improving the manuscript. We uploaded an updated version of the paper and address their comments point-by-point below.
>
> > W1: Fig 1: it is unclear how the patch scores are shown in this figure
>
> In Fig. 1, scores are indicated by means of a translucent grayscale overlay where high scoring patches appear white. This figure motivates that our method can process large images and is able to select salient patches. We have revised the caption of Fig. 1 slightly to make it clearer. The same patch score visualization is used in Fig. 5 (middle) and Fig. 10.
>
>
> > W2: under Eq.1, X_1,...,X_M is not well defined.
>
> Thanks for the tip. We added more details. X itself is defined in the second sentence of Section 2.
>
> > W3: page 2, last paragraph: I suggest the authors add more details here.
>
> The last paragraph of page 2 refers to the data loading options of IPS. We added a detail to the eager loading variant, however we had to condense the information slightly given the page limit.
>
> > W4: under Eq 3, the variables just under Eq 3 need to be discussed in more detail
>
> We have added details to all these variables in the revised manuscript.
>
> > W5: in the paper's prior work, the authors should make a better connection to early approaches for simulating saccades in vision and attention beams.
>
> Thank you for bringing these works to our attention. We integrated some of them, (Culhane &Tsotsos, 1992; Koch & Ullman, 1987 and Burt, 1988), into the related work section (third paragraph) and are happy to take further suggestions in case we missed any important one.
>
> > W6: Appendix A pseudocode: this is not pseudocode, it is actual code. I suggest edits to make it more readable.
>
> Thank you for the tip. We edited the pseudocode to be more readable.
>
> Thank you again for the review. We are happy to take further suggestions.

---

> > ### Comment · Reviewer_jHHu · 2022-11-22
> > **response to author response**
> >
> > The authors have addressed my concerns, so I will leave my ranking of the paper unchanged

---

### Official Review · Reviewer_jpUm · 2022-10-24

**Confidence:** 4
**Correctness:** 4
**Technical Novelty And Significance:** 3
**Empirical Novelty And Significance:** 3
**Recommendation:** 8

**Clarity, Quality, Novelty And Reproducibility:**

The manuscript is easy to read and understand and is of high quality.

The proposed method is novel in its clever re-use of the multi-head cross-attention.

The algo figure in Appendix 1 as well as other details allows for one to reproduce the approach.

**Strength And Weaknesses:**

The manuscript is written quite well and the related work is handled particularly well. The experimental settings are clear and seem to be fair, with many related works being reproduced by the authors themselves. The IPS Transformer clearly performs well while being less sensitive to the choice of M, uses lower GPU RAM, and has competitive inference times.

Though the proposed algorithm is not excessively complex, the concrete steps of how to train IPS Transformer is scattered in the text of Sec. 2. It is therefore difficult to clearly understand the steps involved and which modules are trained when. The paper may benefit from either an algorithm figure in the main paper or a pipeline/workflow diagram figure.

The experimental results and ablation studies look complete.

**Summary Of The Paper:**

This paper proposes a new method of multiple-instance learning from high resolution images, which uses a patch selection scheme that processes available patches in mini-batches, scores them, then aggregates the top-M highest-scoring patches’ embeddings to reach a bag/image-level prediction. The proposed IPS Transformer out-performs state-of-the-art methods on 3 different high resolution image benchmarks. It is shown that IPS Transformer can be less sensitive to the choice of pool size (M), use less memory, while performing similarly to existing methods.

**Summary Of The Review:**

The paper presents a practical yet simple solution to MIL on high-resolution images and experimentally validates its performance on 3 benchmarks. The presentation and analysis are of high quality, warranting the paper’s acceptance.

Edit after rebuttal: I did not previously have major concerns regarding this submission, and after carefully reading the other reviews as well as the authors' diligent response and action, I maintain my rating of 8 as no further concerns exist from my side.

---

> ### Author Response · Authors · 2022-11-15
> **Response to Reviewer jpUm**
>
> We thank the reviewer for their thorough review and positive feedback. We addressed their suggestion for a clearer depiction of the overall workflow by making several changes to the manuscript and in particular Figure 2, which we summarize in the following:
>
> - In the first paragraph of Sect. 2, we specifically mention that the two proposed modules (patch selection and patch aggregation) are executed consecutively, which follows the overall structure of the methods section.
> - We adjusted Fig. 2 by enumerating these two steps: Patch selection is executed first, patch aggregation thereafter.
> - The two modules of Fig. 2 are not separated anymore. Instead, their relations are depicted more clearly: (1) we indicate weight sharing within the cross-attention layer, which is used by both modules, (2) the input to the aggregation module (green box on the bottom right side of the figure) is described as output of IPS, which is now also reflected in the figure caption.
> - The input/output of individual Top-M selections is now shown in Fig. 2.
>
> Thank you again for the review. We are happy to take further suggestions.

---

### Official Review · Reviewer_Ffm2 · 2022-10-25

**Confidence:** 4
**Correctness:** 3
**Technical Novelty And Significance:** 3
**Empirical Novelty And Significance:** 3
**Recommendation:** 6

**Clarity, Quality, Novelty And Reproducibility:**

The paper is presented in a clear way. Figures are very easy and helpful. The novelty is limited. Code is not provided, so reproducibility is a question mark.

**Strength And Weaknesses:**

Strong point:
+ Image patch searching for high-resolution image is an interesting and practical problem.
+ The proposed solution is technically sound.
+ The evaluation results are comprehensive.

Weak points:
- Compared with "Differentiable Patch Selection (DPS)" which also use transformer based image patch selection, the contribution is not that significant.
- The no-gradient mode is confusing to me. Why can't you use the encoder transformer to compute the scores for all the patches, then use some approximation function (e.g., softmax) to estimate the top-k function. This will make the whole DNN differentiable.
- Not sure whether the proposed architecture will bring additional hardware constraint due to the storage of the intermediate results within the DNN. The author should justify that.

**Summary Of The Paper:**

This paper proposes  a simple method, Iterative Patch Selection (IPS), which can select the image patches from a high-resolution image and meet the memory constraint. Two-stage DNN is used to produce the final result, the first part of the DNN is used to select the input patches in autoregressive fashion. The selected patches are then processed by the second part of the DNN to produce the final results. The results indicate that ISP can lead to superior accuracy with minimum level of memory footprint.

**Summary Of The Review:**

Image patch selection for large-scale image is an interesting and practical problem. The paper adopts the transformer based DNN to perform both image patch selection and classification. However, I am a bit concerned about the novelty of the work and the reproducibility of the work, given that no code has been submitted.

---

> ### Author Response · Authors · 2022-11-15
> **Response to Reviewer Ffm2**
>
> We thank the reviewer for their thoughtful review and feedback. We provide a point-by-point response to their concerns and questions below.
>
> > Code is not provided, so reproducibility is a question mark.
>
> We are happy to share that our code is now available as a supplementary zip file. This should allow one to reproduce the Megapixel MNIST experiments for our method. Additional details for reproducing the experiments are provided in the appendix as described in the reproducibility statement. We will release the full source code upon acceptance.
>
> > Compared with "Differentiable Patch Selection (DPS)" which also use transformer based image patch selection, the contribution is not that significant.
>
> DPS is conceptually quite different from IPS. The following table summarizes the differences between these two approaches:
>
> | Criterion     | DPS | IPS     |
> | :---        |    :---   |          :--- |
> | Patch selection input      | Low-resolution image       | High-resolution patches   |
> | Patch selection encoder   | Shallow CNN        | Deep CNN       |
> | Patch selection function   | Differentiable Top-K        | Cross-Attention Top-K       |
> | Patch selection mode   | Gradient mode        | No-gradient mode       |
> | Patch aggregation encoder   | Deep CNN (different to above)        | Deep CNN (same as above)       |
> | Patch aggregation function   | Self-attention + Avg. pool       | Cross-Attention (same as above)
>
> Crucially, DPS encodes all images, albeit in low resolution, through the scorer network in gradient mode. This necessitates storing intermediate activations for backprop, resulting in high memory usage and rendering DPS impractical for very high-resolution settings as discussed in Sect. 4.2.
>
> Furthermore, we find various practical advantages for IPS compared to DPS, as shown in the experiments:
> - Better performance (Table 1 and Table 3)
> - Less sensitive to the choice of M (Table 1 and Table 3)
> - Requires less memory and trains faster (Table 1 and Table 3)
>
> > I am a bit concerned about the novelty of the work
>
> We make two main novel contributions. First, our method enables the training of deep neural networks on arbitrarily large images even on low-cost consumer hardware with limited VRAM. For example, we can train on 10k x 10k pixel images using as little as 1.7 GB of VRAM. Similarly, we can process gigapixel-sized images with sequence lengths of up to 250k (this corresponds to ca. 20% of the size of the ImageNet-1k dataset) using less than 5 GB of VRAM. Showing such large savings is novel and contributes to more accessible AI development.
>
> Second, we observe that a cross-attention based transformer layer is a versatile MIL pooling function under the weighted collective assumption, which has not been shown before to the best of our knowledge. We further show that the attention scores computed by the transformer layer can be used for both patch selection and patch aggregation, and demonstrate competitive performance in different domains and training regimes compared to other well-known baselines.

---

> > ### Author Response · Authors · 2022-11-15
> > **Response to Reviewer Ffm2 (continued)**
> >
> > > The no-gradient mode is confusing to me. Why can't you use the encoder transformer to compute the scores for all the patches, then use some approximation function (e.g., softmax) to estimate the top-k function. This will make the whole DNN differentiable.
> >
> > This would be a valid approach. However, it would increase memory usage and training runtime because a full computational graph would have to be cached while encoding all the patches. We nevertheless implemented the suggested approach as follows:
> >
> > 1. Encode all patches jointly
> > 2. Select the top-k patches according to the attention scores
> > 3. Apply softmax to logit scores of selected patches
> > 4. Aggregate selected patches with scores from 3. and classify
> >
> > This approach requires 15 GB of VRAM for M=100 in Megapixel MNIST, while IPS requires only 1.8 GB for the same setting. The no-gradient mode ensures that no intermediate activations are saved and is hence faster and more memory efficient.
> >
> > We would like to again highlight the necessity for selecting patches in an iterative manner. Even when using the no-gradient mode, encoding all patches at once can be expensive. For example, in Fig. 4 (left), compare I=all vs. I=100 for M=10. When encoding all patches at once, memory requirements are much higher (ca. 5 GB) compared to the iterative setting (< 1GB). Further evidence for this is shown by our TopMIL baseline, which consistently exhibits a higher memory footprint by encoding all patches at once. In some cases, this is too expensive for a single GPU (e.g., experiments with Megapixel MNIST and N=3,481 resulted in out-of-memory errors on an A100 GPU with 40 GB VRAM capacity; in CAMELYON, TopMIL requires 19.8 GB for only 70k patches). With IPS, one can obtain competitive performances in all shown datasets with max. 5 GB.
> >
> > > Not sure whether the proposed architecture will bring additional hardware constraint due to the storage of the intermediate results within the DNN.
> >
> > We have not experienced any additional hardware constraints and designed the method in particular to process very high-resolution images in a fast and memory-efficient way. IPS only caches the top M feature embeddings corresponding to the current memory buffer, and these are comparatively small. Although our proposed method is a 2-stage approach, it is much more memory-efficient and oftentimes faster than 1-stage approaches like CNNs applied to the full image (this is shown in Fig. 3, for example).
> >
> > Thank you again for the review. We are happy to answer further questions.

---

> > > ### Comment · Reviewer_Ffm2 · 2022-12-10
> > > **Feedback**
> > >
> > > The authors provide a very detailed explanation, especially the difference between DPS and IPS. I have updated my scores accordingly.

---

### Official Review · Reviewer_AtFj · 2022-10-26

**Confidence:** 4
**Correctness:** 3
**Technical Novelty And Significance:** 2
**Empirical Novelty And Significance:** 3
**Recommendation:** 6

**Clarity, Quality, Novelty And Reproducibility:**

Clarity - can be improved, specifically novel parts such as learning a scorer and what is fed to the task transformer.
Quality - paper is well written.
Originality of the work - potentially low as the approach is a hand-designed system with little theoretical justification.

**Strength And Weaknesses:**

Strength:

- The problem is very relevant and considers important problem outside of the main focus. Solving CV tasks at high resolution is challenging and authors provide a working (according to results) solution.
- Overall novelty of the paper is in 2 components: (i) iterative patch selection and (ii) patch aggregation.
- The idea of using transformer attention for scoring that is learned without gradient is novel and seem to work fine.
- The related work section is quite broad and gives a nice overview for the person not in the field.

Weaknesses/questions:

1) Significance of the method and novelty. The approach seem to be hand designed and heuristic. We don't see much of the theory to back it up. The contribution of (i) iterative patch selection can be considered as a straightforward and not novel. Patch aggregation with the idea of scoring patches is also not groundbreaking. The way patch scores are training is an in interesting point.
2) The explanation of how attention scorer is trained is a bit vague. Paper states multiple times that there is no gradient propagation (no-gradient mode), but then it is not clear how to train the scorer. There is a second network that aggregates patches, but it seem to take patch + positional encoding meaning previous scores are not needed. If both blocks share the same attention then we can train only the second one, and use attention scores for the selector. A more detailed explanation of the novel part is needed.
3) The implementation uses a relatively heavy network resnet50 to embed patches. This network by itself can potentially solve the problem if information from all patches is aggregated by some way of pooling (max for example). Probably learning an embedder will be useful for the work.



**Summary Of The Paper:**

The problem of solving computer vision tasks with neural networks on large images is considered. Authors propose a method that splits an image into patches and then applying greedy iterative approach (IPS iterative patch selection) to select the representative ones. Selection is performed by creating a buffer and iteratively updating it by scoring them. Final buffer is then transformed via a convex combination of embeddings. The approach is more heursitic than theoretically justified. Results are demonstrated on multiple dataset where high resolution matters, the method is compared with multiple others.

**Summary Of The Review:**

Paper tackles interesting problem of working with high resolution images. The solution is a system that selects informative patches and processes them with transformer. The novelty of the method is on the weak side and some components that can be considered interesting to the community are not well explained. Given little novelty and no significant theoretical justification of the method my recommendation is top reject the paper.


------Post rebuttal------
Thank authors for addressing raised points. The way attention module is trained without gradients is more clear now and updates to the figure are great. Reading other reviews it seems they are more positive. On my end, paper has only empirical contribution is it is not groundbreaking, however, given other reviews I will increase my score to 6 and will not object if the paper is accepted.

---

> ### Author Response · Authors · 2022-11-15
> **Response to Reviewer AtFj**
>
> We thank the reviewer for their thoughtful review and feedback. We provide a point-by-point response to their concerns and questions below.
>
> > The approach is more heursitic than theoretically justified.
>
> Our work does not attempt to make theoretical contributions. Instead, we focus on solving an important problem – the processing of arbitrarily large images under tight hardware constraints – which is useful for many important real-world applications, as is demonstrated for datasets relating to autonomous driving and computer-aided diagnosis. As such, we attempt to make significant empirical contributions with respect to computational efficiency, which we believe are important to the broader community.
>
> However, we agree that it is important to justify design decisions, which ought to be chosen in a well-informed manner. Our approach can be placed in the Multiple Instance Learning (MIL) literature as described in the first paragraph of Sect. 3. In particular, it follows the weighted collective MIL assumption, which expresses that all instances contribute to the bag representation, albeit to varying degrees according to the attention scores. In IPS, contributions of instances are restricted to selected patches only, and weights of non-selected patches are thus set to 0, which is key to computational efficiency.
>
> Moreover, the IPS Transformer can be viewed as an embedding-level MIL approach where a bag-level representation is obtained prior to classification. This is opposed to less flexible instance-level approaches that classify each instance and aggregate their scores thereafter.
>
> Another property of the cross-attention formulation is that it constitutes a permutation-invariant MIL pooling function (similar to Ilse et al., 2018), i.e. patches are scored independently of each other. This implies that IPS selects the same patches independent of the composition of intermediate memory buffers and thus the order in which IPS processes the patches does not matter.
>
> > The contribution of (i) iterative patch selection can be considered as a straightforward and not novel. Patch aggregation with the idea of scoring patches is also not groundbreaking. The way patch scores are training is an in interesting point.
>
> The method is deliberately designed to be easy to understand and adopt. While in general score-based selection is not novel, our specific approach is. Specific novel elements of our IPS Transformer are:
>
> - Employing a cross-attention layer as MIL pooling function used for both patch selection and patch aggregation.
> - Selecting patches in no-gradient mode to ensure fast and memory-efficient selection, as well as in evaluation mode to ensure deterministic patch selection behavior when using BatchNorm and Dropout.
> - Three data loading options for different use cases that trade off memory and runtime efficiency in IPS, as demonstrated in our ablation study.
>
> > Paper states multiple times that there is no gradient propagation (no-gradient mode), but then it is not clear how to train the scorer.
>
> It is correct that IPS runs in no-gradient mode and thus cannot train the parameters of the cross-attention layer (scorer). However, by sharing the parameters with the trainable patch aggregation module, it can still learn to select relevant patches. This is described in the last three sentences of Sect. 2.2. We have also updated Fig. 2 to make this connection clearer.
>
> > There is a second network that aggregates patches, but it seem to take patch + positional encoding meaning previous scores are not needed. If both blocks share the same attention then we can train only the second one, and use attention scores for the selector.
>
> Indeed, both patch selection and patch aggregation modules share the same cross-attention layer, but for different purposes. In the patch selection phase, the scores are used to select the top-scoring patches. Only these patches are subsequently re-embedded and the resulting feature vectors are used as input to the aggregation module, in which the cross-attention layer scores them again for training. So, we need the patch scores from the IPS stage to reduce the number of patches to be processed in the subsequent and more expensive training stage to save memory. A simple reuse of scores, as opposed to a full recomputation in the patch aggregation module, is not feasible as then IPS will need to run in gradient mode leading to out-of-memory errors in high resolution images.

---

> > ### Author Response · Authors · 2022-11-15
> > **Response to Reviewer AtFj (continued)**
> >
> > > The implementation uses a relatively heavy network resnet50 to embed patches. This network by itself can potentially solve the problem if information from all patches is aggregated by some way of pooling (max for example).
> >
> > We use a simplified ResNet-18 (2 ResNet blocks) trained from scratch for Megapixel MNIST, an ImageNet-1k pre-trained ResNet-18 for traffic sign recognition, and a ResNet-50 trained in a self-supervised manner for feature extraction in CAMELYON-16 (similar to Dehaene et al., 2020).
> >
> > Our TopMIL baseline is similar to the suggested approach, i.e. patches are classified individually and the top M patch predictions are aggregated. This approach has several drawbacks, which are summarized below:
> >
> > - Irrelevant patches might be included, which can harm performance. For example, in traffic sign recognition (Sect. 4.1), aggregating 10 patches performs worse than aggregating only 2 patches. This means that TopMIL requires knowledge about the task, especially about the signal-to-noise ratio in the data.
> > - TopMIL requires more memory because the patches are scored jointly. For example, in CAMELYON-16 (Sect. 4.2), TopMIL requires 19.8 GB, while IPS requires only up to 4.7 GB.
> > - TopMIL does not integrate positional information well because it classifies patches directly without computing an image-level embedding (as opposed to IPS). As a result, TopMIL performs much worse than IPS on Megapixel MNIST (Sect. 4.3), when a position context (task ‘top’) is required.
> >
> >
> > Thank you again for your review. We are happy to answer further questions.

---

### Author Response · Authors · 2022-11-15
**General response**

We would like to thank the reviewers for their thoughtful reviews and positive feedback. We are pleased that the reviewers appreciated that we address a challenging and important problem interesting to the community (Reviewers Ffm2, AtFj, jHHu), the novelty of our method (Reviewers AtFj, jpUm), the comprehensive evaluation and good results (Reviewers Ffm2, AtFj, jpUm), as well as the clear presentation of the paper including related works (Reviewers jpUm, jHHu, AtFj).

In response to the feedback, we uploaded supplementary source code and revised the manuscript as follows:

- We adjusted Fig. 2 and added details to better illustrate the overall workflow.
- We added more details to the method subsections 2.1 and 2.2.
- We integrated early attention approaches in the related works.
- We adjusted the pseudocode in Appendix A to improve readability.

Detailed responses to individual reviews are given below.

---

### Decision · Program_Chairs · 2023-01-20

**Decision:**

Accept: poster

**Justification For Why Not Higher Score:**

The paper presents a simple and practical method that will help researchers train models on high-resolution images. At present, use of high-resolution images for training is a bit of a niche use case though so I believe this work is best presented to a targeted audience interested in that niche use case via a poster presentation.

**Justification For Why Not Lower Score:**

The paper presents a simple and practical method that will help researchers train models on high-resolution images. The method will surely be of interest to (part of) the ICLR audience.

**Metareview: Summary, Strengths And Weaknesses:**

The paper presents a strategy for training deep networks on very large input images by selecting only salient parts of those images and training on those parts only. This strategy substantially reduces how much memory is required for training, hence solving an important practical obstacle for the use of deep networks on high-resolution images. The paper presents experiments on high-resolution traffic images and medical images, validating the efficacy of the approach relative to prior work.

Some reviewers question the paper’s novelty because the strategy presented is relatively simple. The AC disagrees with this view: a simple method that works well is much more likely to be widely adopted than a complex method.

The manuscript’s presentation of the IPS method is confusing at times, as pointed out by multiple reviewers. The authors are encouraged to improve the presentation in the camera-ready version of the paper.

**Note From Pc:**

if the above contains the word "oral" or "spotlight" please see: "oral" presentation means -> notable-top-5% and "spotlight" means -> notable-top-25%. As stated in our emails, we are disassociating presentation type from AC recommendations